# RETRACTED: Sinapic Acid Attenuate Liver Injury by Modulating Antioxidant Activity and Inflammatory Cytokines in Thioacetamide-Induced Liver Cirrhosis in Rats

**DOI:** 10.3390/biomedicines11051447

**Published:** 2023-05-15

**Authors:** Ahmed A. J. Jabbar, Zaenah Zuhair Alamri, Mahmood Ameen Abdulla, Ahmed S. AlRashdi, Soran Kayfi Najmaldin, Mustafa AbdulMonam Zainel

**Affiliations:** 1Department of Medical Laboratory Technology, Erbil Technical Health and Medical College, Erbil Polytechnic University, Erbil 44001, Iraq; 2Department of Biological Sciences, Faculty of Science, University of Jeddah, Jeddah 23218, Saudi Arabia; zzalamri@uj.edu.sa; 3Department of Medical Microbiology, College of Sciences, Cihan University-Erbil, Erbil 44001, Iraq; mahmood.ameen@cihanuniversity.edu.iq; 4Central Public Health Laboratories, Ministry of Health, P.O. Box 2294, Muscat 111, Oman; ahmed.alrashdi2@moh.gov.om; 5Department of Medical Analysis, Faculty of Applied Science, Tishk International University, Erbil 44001, Iraq; soran.kayfi@tiu.edu.iq; 6Department of Pharmacy, College of Pharmacy, Knowledge University, Erbil 44001, Iraq; mustafa.zainel@knu.edu.iq

**Keywords:** sinapic acid, liver cirrhosis, TAA, histology, immunohistochemistry

## Abstract

Sinapic acid (SA) is a natural pharmacological active compound found in berries, nuts, and cereals. The current study aimed to investigate the protective effects of SA against thioacetamide (TAA) fibrosis in rats by histopathological and immunohistochemical assays. The albino rats (30) were randomly divided into five groups (G). G1 was injected with distilled water 3 times/week and fed orally daily with 10% Tween 20 for two months. G2–5 were injected with 200 mg/kg TAA three times weekly for two months and fed with 10% Tween 20, 50 mg/kg silymarin, 20, and 40 mg/kg of SA daily for 2 months, respectively. The results showed that rats treated with SA had fewer hepatocyte injuries with lower liver index (serum bilirubin, total protein, albumin, and liver enzymes (ALP, ALT, and AST) and were similar to that of control and silymarin-treated rats. Acute toxicity for 2 and 4 g/kg SA showed to be safe without any toxic signs in treated rats. Macroscopic examination showed that hepatotoxic liver had an irregular, rough surface with micro and macro nodules and histopathology expressed by Hematoxylin and Eosin, and Masson Trichrome revealed severe inflammation and infiltration of focal necrosis, fibrosis, lymphocytes, and proliferation bile duct. In contrast, rats fed with SA had significantly lower TAA toxicity in gross and histology and liver tissues as presented by less liver tissue disruption, lesser fibrosis, and minimum in filtered hepatocytes. Immunohistochemistry of rats receiving SA showed significant up-regulation of HSP 70% and down-regulation of alpha-smooth muscle actin (α-SMA) protein expression compared to positive control rats. The homogenized liver tissues showed a notable rise in the antioxidant enzymes (SOD and CAT) actions with significantly lower malondialdehyde (MDA) levels compared to that of the positive control group. Furthermore, the SA-treated rats had significantly lower TNF-a, IL-6, and higher IL-10 levels than the positive control rats. Thus, the findings suggest SA as a hepatoprotective compound due to its inhibitory effects on fibrosis, hepatotoxicity, liver cell proliferation, up-regulation of HSP 70, and downregulation of α-SMA expression, inhibiting lipid peroxidation (MDA), while retaining the liver index and antioxidant enzymes to normal.

## 1. Introduction

There is an upsurge in liver fibrosis prevalence worldwide, with an estimated 1.5 billion cases according to the latest statistics reported in 2020, indicating a 13% increase since 2000 [1]. Cirrhosis has become the 11th leading cause of death (2.2%) and 15th leading cause of morbidity (1.5%) worldwide in 2016 [2]. Liver disease, as a chronic health condition, may be caused by hepatitis B virus infection, the hepatotoxic effects of chemicals such as alcohol, metabolic associated fatty liver disease, medications, and environmental pollutants. Natural products and phytochemicals have been relayed on as alternative medicine for centuries for liver disease in various populations. The scientific literature has shown many studies revealing beneficial medicinal effects of numerous medicinal herbs in protecting the liver, contrary to thioacetamide (TAA)-influenced damage in laboratory animals [3,4]. Liver cirrhosis stimulated by thioacetamide (TAA) is a laboratory-known model for causing hepatocyte injury, producing nodules, and liver fibrosis same as liver fibrosis in humans [5]. TAA has the potential to increase the production of reactive oxygen species and free radicals while reducing the glutathione (GSH) and antioxidant enzymes (SOD and CAT) in the hepatocytes leading to lipid peroxidation, oxidative stress, and cell necrosis [6,7].

Sinapic acid (family: phenylpropanoid) is a natural derivative of hydroxycinnamic acid that has been known for composing various bioactivities, anti-inflammatory, antiradical, and antitumor actions [8,9]. Recent Studies revealed that SA and its derivatives (syringic acid) exhibit significant anti-inflammatory and anti-radical potentials in chemical-induced kidney and liver injuries in rats [8,10]. Furthermore, researchers have reported that SA has the potential to attenuate acute DOX-induced cardiotoxicity by lowering inflammation in rats [11]. Studies have revealed that SA significantly revoked hypertension by lowering reactive oxygen species and reducing tissue fibrosis [12]. The efficacy of SA as a lipid-lowering and antioxidant agent has been reported in the starting stages of myocardial injury [13,14]. However, until now, histopathological studies on the hepatoprotective of SA in TAA-induced liver cirrhosis have not been reported.

Liver cirrhosis has been known as an outcome of fibrogenesis progression with different routes of fibrotic formation, correlated with the main cause of the fibrosis and based on the source of fibrogenic cells [15]. Biliary fibrosis is identified by the generation of reactive bile ductules and periductular myofibroblast-like cells at the interface of portal-parenchyma with the direction of portals resulting in the production of portal-portal septa enclosing liver nodules. In case of the chronic viral hepatitis, the portal-central (vein) linkage with necrosis results in the production of portal-central septa and quick reordering of the vascular connections with the portal system causing portal hypertension [16]. The central type of fibrogenic initiation considered a secondary problem to the venous outflow and is recognized by the formation of central to central septa and reversed lobulation. The pericellular/perisinusoidal usually results from Nonalcoholic fatty liver disease (NAFLD), Nonalcoholic steatohepatitis (NASH), and alcohol overuse, in which the leaked fibrillary matrix is accumulated around the sinusoids (capillarization) and hepatocytes. The various routes of fibrosis differentiation are linked with four main causes, an area where tissue damage occurred, the intensity of pro-fibrogenic cause, common pathways of pro-fibrogenic, and the source of the pro-fibrogenic myofibroblast [15].

Liver injury correlated with hepatic inflammation is facilitated mainly by the proinflammatory (TNF-a, cytokines tumor necrosis factor and IL-1 β, interleukin-1 β) [17]. More Studies also reported that pro-inflammatory cytokines (IL-1 α and β) and (TNF-α) are major contributors in progressive stages of hepatocyte diseases through direct involvement in many hepatocellular processes, acute phase protein synthesis, metabolism of lipid, cholestasis, and fibrosis stages [18,19]. The activity of these cytokines is regulated by the transcription factor nuclear factor-kB (NF-kB) [20]. Furthermore, TNF- α and IL-1 β have the potential to stimulate the NF-kB pathway, thus providing a continuous auto-regulatory cycle that can multiply and increase the period of an inflammatory response [21]. Thus, cytokine enzymes (TNF-α, IL-6, and IL-10) have gained more attention as a target to prevent or treat inflammatory-related diseases such as liver cirrhosis [22].

Searching for new natural compounds as alternative medicine effective against liver injury has gained more attention from researchers in recent years due to the drawbacks related to chemically synthetic drugs [23,24]. Thus, scientists have explored and shown the therapeutic efficacy of numerous plant-based compounds including SA against many health issues (renal, liver, diabetic, and heart diseases) [10,12,14]. A systematic search showed a lack of any concise experimental analysis explaining the bio-mechanisms behind the protective role of SA against liver cirrhosis. Therefore, the current study aims to investigate the hepatoprotection efficiency of SA in TAA-induced liver cirrhosis in rats for the first time.

## 2. Material and Methods

### 2.1. Chemical and Standard Drug

The thioacetamide (TAA) was purchased from Sigma-Aldrich (Buchs, Switzerland) and then liquefied in 10% Tween 20 and mixed for complete dissolution. Meanwhile, 200 mg/kg body mass intraperitoneal injection (i.p.i.) was delivered to rats 3 times weekly for two months. The TAA induced vicissitudes along with biological changes in morphology and structures similar to that of liver cirrhosis in humans [25].

Silymarin as a standard medication (International Laboratory, San Francisco, CA, USA) was utilized as a reference drug for liver cirrhotic rats. Silymarin is a flavonolignan that has been used extensively as a standard drug against hepatocyte damage. Silymarin has shown significant efficiency in curbing the plentiful liver damage characterized by deteriorating necrosis and tissue fibrosis [26]. Thus, researchers have relied on silymarin as a standard or reference medication as a liver protectant against TAA-induced liver cirrhosis in rats [27,28,29,30,31]. Silymarin was dissolved in 10% Tween 20 and delivered to rats in 50 mg/kg dosage via oral gavage technique [3].

### 2.2. Sinapic Acid (SA)

The SA was purchased from Sigma-Aldrich Chemical Co., (Saint Louis, MO, USA). The dissolution of SA was made in 10% Tween 20 and then, intragastrically delivered to rats in 20 and 40 mg/kg dosages [32].

### 2.3. Acute Toxicity Study and Experimental Animals

The Sprague Dawley rats (18 males and 18 females) 6–7 weeks old, weighed 180–210 g were provided by the Experimental Animal House, Cihan University-Erbil. The rats were receiving a standard diet (rat pellets) and tap water (ad libitum) and were caged in separate cages (coprophagia avoided via wide-mesh wire at the bottom). The rats were caged for 7 days for adaptation and then, the acute toxicity test was performed to find a safe dosage of SA. The rats were separated randomly into three groups (each with 6 males and 6 females); vehicle (10% Tween 20, 5 mL/kg), 2 g/kg, and 4 g/kg of SA. Before the dosage was given, overnight fasting was applied to all experimental rats (free water access). Food was also taken away for another 3 to 4 h after the dosage was given. The animal’s observation began at 24–48 h after the dosing of SA for any clinical or toxic signs or mortality [33]. The record continues for the straight 14 days. The animals were then sacrificed by overdosing on anesthesia [(ketamine (30 mg/kg, 100 mg/mL) and xylazine (3 mg/kg, 100 mg/mL)]. on the 15th day. The collected blood samples from intracardial puncture were centrifuged and the gained serum was biochemically analyzed. The liver and the kidney were investigated histologically based on standard methods [34].

### 2.4. Experimental Animals for Hepatoprotective Activity

Sprague Dawley rats were provided by the Animal House Unit, Cihan University-Erbil. Rats (180–200 g) were caged individually in wide-mesh wire cells throughout the experimental period at a suitable condition (approximate moisture 55–65%, 25 ± 2 °C temperature, and 12-h exposure light/dark cycle). All the rats received tape water and a standard pellet. Sprague Dawley male rats (30) were separated into five groups with six rats in each group following the previously explained technique [31]. The control group (G1), was addressed with distilled water (5 mL/kg) i.p.i. 3 times a week and a daily oral dosage of 10% Tween 20 (5 mL/kg) for two months. The positive control rats (G2) received i.p.i. (200 mg/kg) of TAA three times a week and a daily oral dosage of 10% Tween 20 (5 mL/kg) for two months. The reference drug (G3) received TAA (200 mg/kg) i.p.i. three times a week and daily oral ingestion of Silymarin (50 mg/kg) for two months. The SA (G4 and G5) groups were given TAA (200 mg/kg) i.p.i. three times a week and daily oral ingestion of SA 20 mg/kg (G4) and 40 mg/kg (G5) for two months [35].

After the 2-month experimental period, all rats were fasted for 24 h and then given anesthesia (ketamine and xylazine) at 30 mg/kg (100 mg/mL) dosage. Blood samples were collected from intracardial puncture and the separated serum was analyzed for liver biochemical parameters [36].

### 2.5. Biochemical Parameters (Liver Function Test)

The blood samples were collected using clot activator tubes and centrifuged for 15 min at 2500 rpm to obtain serum. The liver enzymes, alanine aminotransferase (ALT), aspartate aminotransferase (AST), alkaline phosphatase (ALP), and the total bilirubin, and total protein in addition to albumin were investigated using spectrophotometry technique [22].

### 2.6. Macroscopic Views of Liver

Liver tissue analysis was performed by exposing the abdominal and thoracic cavities of the rats. The liver sections revealed significant proof of pathological changes presented by clear microscopic views. Furthermore, organs besides the liver were also experienced severe gross lesions, but not included in the present study. The collected livers were washed individually in cold saline and screened grossly for possible pathological changes via microscopic observation [37].

### 2.7. Histopathology of Liver Tissue

The collected livers were cleaned with cold saline, sliced in 2 cm cubic, and fixed using 10% phosphate and buffered formalin. The tissue processing and specimens were made via the Leica machine (Wetzlar, Germany) and then deepen in paraffin. Small slices (5 µm thickness) were routinely stained by hematoxylin and Eosin (H&E). Furthermore, the Masson trichrome stain was also utilized [28]. The slide observation was made via a Nikon microscope (Y-THS, Tokyo, Japan) and different tissue areas were captured [27].

### 2.8. Immunohistochemistry Investigation

The HSP % and α-SMA staining procedures were performed via Poly-L-lysine-handling glass slides. Hepatic tissues (5 μm in thickness) were colored (immunohistochemical staining) based on producers^,^ instructions (Santa Cruz, CA, USA) for using Animal Research Kit. The principle of the procedure began with the blocking of endogenous peroxidase enzyme via hydrogen peroxide sodium azide (0.03%) for 3 min. The obtained liver tissues were washed (buffer) and then stained with HSP 70 (1:100) and d α-SMA (1:25) for fifteen minutes. The stained slides were transferred into a buffer bath. After that, tissue sections were incubated with streptavidin-HRP for fifteen minutes. The slides were then incubated with Diaminobenzidine-substrate chromagen (7 min) followed by washing and counterstaining by hematoxyline (5 s). The slides were deepened in an ammonia container (0.037 M/L) ten times, and antigens were retrieved via microwave boiling in sodium citrate buffer (10 mM). The immunohistochemical stain-positive cells were detected via microscope observations [22,38]. Then, the percent calculation of positive cells was determined.

### 2.9. Liver Tissue Homogenate for Endogenous (CAT, SOD) Enzymes and Oxidative Stress (MDA)

The tissue samples were washed in neutral ice-cold phosphate-buffered saline 10% (*w*/*v*). The homogenate liver tissues were prepared via Teflon homogenizer (performed on ice), and then centrifuged at 4500 rpm for 15 min at 4 °C excluding cell debris. The collected supernatant was examined for antioxidant actions by using superoxide dismutase (SOD) and catalase (CAT) kits (Cayman Chemical Company, Ann Arbor, MI, USA) [39]. Malondialdehyde (MDA), as a lipid peroxidation biomarker, was also assessed to analyze the level of oxidative stress. Thus, the MDA assay kits were purchased and the reacted amount of thiobarbituric acid was determined [40] (TBARS, Cayman Chemical Company).

### 2.10. Estimation of Inflammatory Cytokines

TNF-α, IL-6, and IL-10 were detected using a marketable ELISA kit (Cusabio Biotech Co., Wuhan, China). Briefly, the centrifugation of the liver tissue homogenates was made at 3000× *g* for 15 min and the supernatant was analyzed for the cytokine contents via a commercial enzyme-linked immunosorbent kit. The examination followed the same technique provided in the constructor’s experiment presented in the rat ELISA Kit for TNF-α (109331), IL-6 (84597), and IL-10 (84236) enzymes. Cytokine levels were assessed utilizing standard recombinant purified cytokines [36].

### 2.11. Statistical Analysis of Data

The current data were analyzed using One-way ANOVA, Tukey post hoc test, and SPSS software (version 24), IBM Corp., Armonk, NY, USA. The accuracy of the obtained data was ensured by repeating the process multiple times. The present figures were designed using Graph Pad Prism (version 9.0), GraphPad Software, Inc., San Diego, CA, USA.

## 3. Results

### 3.1. Acute Toxicity Study

Rats treated with 2 and 4 g/kg of SA were observed for 2 weeks. The rats remained active and no clear toxic signs or poisonousness were observed. Furthermore, no changes were seen in behavior, body mass, or gross during or after the experiment with zero mortality at both dosages during or after 14 days of the procedure. Biochemical and histopathological examinations showed non-signification changes among the SA-treated rats in comparison to that of the control group (Table 1 and Figure 1). Therefore, SA was found to be safe at both dosages.

The data biochemical test results of the kidney showed that rats treated with SA (2 and 4 g/kg) had similar values to that of negative control rats with no significant changes based on the statistics shown in (B) in Table 1.

### 3.2. Effect of SA in TAA-Induced Liver Cirrhosis

#### 3.2.1. Body Heaviness, Liver Mass, and Liver Index

The positive control rats (TAA control) had significantly decreased body weight and increased bulky liver weight compared to the control group. Rats receiving silymarin or SA had significantly higher body mass and lower liver bulks (Table 2).

#### 3.2.2. Gross Appearance of Liver

The gross morphology of liver tissues in all groups was examined as shown in Figure 2. The results revealed that the control rats had smooth liver surfaces with regular lobs (Figure 2, gross (A)). Positive control rats (B) had an irregular liver surface with several macro and micronodules (Figure 2, gross (B)). The TAA + silymarin-addressed rats had a smooth liver surface that was comparable to that of the control group (Figure 2, gross (C)). TAA + SA 20 mg/kg or TAA + SA 40 mg/kg treated rats had nearly smooth liver surfaces and almost the same liver structure and shape as the control group (Figure 2, gross (D, E)).

### 3.3. Histopathological Investigation of Liver Sections

The liver tissues were stained with hematoxylin and eosin for observing any histopathological modification as presented in Figure 2 (H & E). Microscope views of liver tissues from the control group showed typical cell structure, maintained cytoplasm, obvious nucleus, and nucleolus with distinct epithelial linings of hepatocytes divided by sinusoidal capillaries and central vein (Figure 2, H & E (A)). Rats treated with TAA had liver tissues with irregular hepatocyte structures because of numerous reforming nodules. Furthermore, liver tissue was separated by fibrous septa elongated from the central vein to the portal area. Hepatocytes appeared with deep injury, necrosis, bile duct propagation, congested central vein, and deformations in the monocytes and granulocytes because of increased inflammation (Figure 2, H & E stain (B)).

Silymarin + TAA, 20 and 40 mg/kg of SA + TAA groups were shown relative protective effects from hepatocyte injury induced by TAA. The liver cell components revealed a decreased level of tissue damage with a slight fibrotic septum.

The liver tissues in these groups appeared with fewer lymphocytes disruption. Moreover, the histopathological results revealed remarkable regenerative parenchymal nodules accompanied by numerous fibrous tissue and normal fat cell growth, Kupffer cells, and bile ducts (Figure 2, H & E (C–E)). To investigate liver fibrosis, the tissue sections were stained with Masson’s trichrome. Results showed the absence of deposited collagen in the liver of the control group (Figure 2, MT (A)). Rates addressed with 200 mg/kg TAA had regenerative bile duct with significant dense fiber septa and numerous collagen fiber accumulations near the congested central vein, indicating severe fibrosis in the hepatic tissues (Figure 2, MT (B)). The silymarin, 20 mg/kg, or 40 mg/kg of SA-treated rats had significantly lower rates of fibrous septa and tissue nodules. Moreover, these groups had normal homologous collagen indicating the hepatoprotective role of SA (Figure 2, MT (C–E)).

Results of collagen leakage are presented as scores in different experimental and controls. The highest score of collagen deposition was recorded for positive control rats, indicating the severe fibrosis rate induced by TAA. Rats treated with SA have shown reduced scores of collagen deposition in liver tissues stained with MT, indicating effective protection of this compound against TAA-induced liver fibrosis (Figure 3).

### 3.4. Immunohistochemical Stain of Hepatocyte

The immunohistochemical effect of SA against TAA-induced liver injury was investigated by evaluating the HSP 70 protein expression in the liver parenchyma using an anti-HSP antibody (Figure 4). Control rats had no HSP 70 expression, meaning no renewal cells were found. In contrast, hepatocytes from the TAA-treated rats showed downregulated HSP 70 expression and higher mitotic rate, indicating severe proliferation and hepatocyte injury liver caused by TAA.

Livers from rats treated with 20 mg/kg, 40 mg/kg SA, or silymarin showed significantly higher cell renewal up-regulation than that of the TAA control down-regulation, as presented by intense HSP 70 expression and a maintained lower mitotic index.

The hepatoprotective effect of SA in TAA-induced liver injury in rats was also investigated by immunohistochemical staining of α-SMA protein expression in the liver parenchyma using typical antibodies. Control rats (A) had lowered α-SMA staining, indicating no regenerative hepatocytes (Figure 5). In contrast, the positive control (B) showed significantly increased α-SMA expression with an increased rate of liver fibrosis and mitotic index, indicating proliferation to the regeneration of liver cell injury induced by TAA. Almost the same results were found in silymarin-treated rats (C) and SA 20 mg/kg treated rats (D) by the significant revitalization of hepatic cells and resistance of hepatic fibrosis in comparison to that of the TAA-treated positive control rats, as presented by moderate expression of α-SMA stain and lower mitotic index. Whereas the SA 40 mg/kg treated rats (E) showed mild to moderate α-SMA staining appearance within the liver cells with a notable decline in the mitotic index but lower than that of the C and D groups. These outcomes scientifically back up the hepatoprotection efficacy of SA by resisting liver fibrosis and lowering hepatocyte propagation.

### 3.5. Effects of SA on Endogenous Antioxidant Enzymes in TAA-Induced Liver Cirrhosis in Rats

The results have shown that TAA-treated rats (G2) had significantly decreased antioxidant enzymes (SOD and CAT) in comparison to the control group (Figure 6). The SA treatment significantly retained the depletion of endogenous antioxidants (SOD and CAT) to normal values (Figure 6a,b). Rats addressed with SA 40 mg/kg (G5) had statistically higher values of SOD and CAT than that of negative controls. Moreover, the SOD values were very comparable between G3, G4, and G5. The lipid peroxidation was estimated by evaluating the MDA level, which was significantly lower in SA 20 mg/kg (G4) and 40 mg/kg (G5) treated rats than that of hepatotoxic groups, respectively. Non-significant changes were found in the MDA levels between the G1, G3, G4, and G5 groups (Figure 6c). The silymarin-treated rats (G3) had statistically different SOD (*p* < 0.0001) and CAT (*p* < 0.05) values than that of the negative control (G1).

### 3.6. Effect of SA on Cytokines (TNF-α, IL-6, and IL-10) in TAA-Induced Liver Cirrhosis in Rats

The positive control rats (G2) had significantly increased TNF-α and IL-6 values, and decreased IL-10, indicating severe immunity status because of severe inflammation caused by TAA injection (G2). In contrast, the Silymarin or SA treatment led to significant immune-enhancement in the liver tissue homogenate presented by reduced TNF-α (Figure 7a), IL-6 values (Figure 7b), and uprising the IL-10 levels (Figure 7c).

### 3.7. Effect of SA on Liver Biochemistry in TAA-Induced Liver Cirrhosis

The liver function parameters (ALT, AST, ALP, and bilirubin) were notably increased significantly in the positive group (G2) in comparison to silymarin (G3) or SA-treated rats (G4 and G5). While, the albumin and total protein were found significantly lower in the G2 group than in that of the G3, G4, and G5 groups (Table 3). The liver parameters were significantly downregulated in rats receiving SA or Silymarin.

The TAA injection in the G3 group significantly upraised (*p* < 0.05) liver enzyme biomarkers, as shown in Table 3. Furthermore, the positive control rats showed lower total protein and albumin than the control group, indicating severe liver damage. Rats treated with SA (G4 and G5) or silymarin (G3) had significantly lower liver enzymes and higher total protein and albumin levels than that positive control rats (G2). Thus, the SA suppressed the liver fibrosis and the hepatotoxicity of TAA by upraising the liver enzymes. The SA at 20 mg/kg dosage was more effective in recovering the liver enzymes in TAA-induced liver cirrhotic rats than that of 40 mg/kg dosage.

## 4. Discussion

The current study showed that positive control rats experienced significant up-regulation of the liver biomarkers (ALT, ALP, AST, and bilirubin). Researchers have reportedly confirmed the TAA efficacy in disturbing the biochemical status of the liver could lead to severe liver damage and hepatic dysfunction. Furthermore, TAA ingestion significantly raised the liver bioactivities and bilirubin values, which is following the previously published studies, specifying the TAA interactions with RNA and nucleus initiating extracellular hepatic damage leading to increased liver activities and thus, more liver enzymes leakages into the circulated blood [4,6,31]. These liver function parameters were retrieved to almost normal values in rat treated SA group. Similarly, numerous studies have shown the efficacy of medicinal plants or their natural compounds in lower liver enzymes and the rate of bilirubin production [4,37,41]. The hepatoprotective action of SA could be linked with its inhibitory action against hepatocellular leakage and fibrotic injuries surrounding the hepatic cells. Researchers have shown SA as a hepatic protectant against CCl4-induced inflammation and have linked this bioactivity with its ability to reduce oxidative stress and activation of NF-κB, p65, and proinflammatory cytokine signaling pathways [24,42].

The present outcomes have shown that the serum albumin and total protein were notably decreased in positive control rats induced by TAA. In contrast, rats treated with silymarin or SA recovered these abnormalities to the near standard range. Our results are following previous investigations reported by several researchers stating that oral gavage of silymarin or phytochemicals has normalized the albumin and protein in cirrhotic rats [4,37].

The outcomes have shown significantly lower body weight and higher liver weight (hepatomegaly) in positive control rats than that in control rats. This liver/body weight change could be linked with adipocyte accumulation and hepatocyte deterioration. Similarly, studies have reported the TAA efficacy in changing the weightage of the body and bulky liver formation in various rat trials [4,42,43]. These body/liver weightage modulations were almost normalized in SA-treated rats, which could be correlated with the dyslipidemia [44], anti-inflammatory [45], and antioxidant [46] bioactivities of SA as reported by previous researchers. Similarly, researchers have reported the SA efficacy in restoring the body and liver weightage to normal in dimethylnitrosamine (DMN)-induced chronic liver damage in rats [47].

The present data revealed decreased deposited collagen in SA-treated rats based on the histopathological evaluation of liver tissues stained with Masson’s trichrome dye. Accordingly, scientists have shown the improving effect of natural products in curbing the increased collagen decomposition in cirrhotic rats induced by TAA [4].

The current study has shown that the TAA inoculation in rats caused severe hepatotoxic damage and the SA treatment revealed the significant suppressing effect of this liver damage as shown through histopathological (H & E), reduced Masson’s Trichrome stains, and up-regulated the HSP 70 immunostain, indicating its hepatoprotection actions. Similarly, studies have shown the SA potentials in the positive modulation of the HSP 70 protein expression in different rat experiments [9,48]

The endogenous antioxidant enzymes in the homogenized hepatic tissues are considered significant indicators of oxidative stress-related liver damage [49]. The present data revealed that rats treated with only TAA had significantly lower liver antioxidant enzymes (SOD and CAT) than that found in control rats. In contrast, the SA-treated rats had significantly higher SOD and CAT values in their tissue homogenates than that of positive control rats, revealing the antioxidant actions of SA against free radicals-related liver injury. Similarly, researchers have shown the free radical quenching potentials of SA in different rat trials [14,50]

The MDA as an important lipid peroxidation indicator has been relying on scientists to allocate the level of oxidative stress [22,34]. The positive control rats showed increased MDA levels, indicating severe lipid peroxidation and elevated oxidative hepatocyte damage and possibly a reason behind the reduced liver antioxidants and hepatic fibrosis. This free radical damage and TAA intoxication were shown significantly lesser in SA-treated rats as the MDA values were much lower than that of positive control rats. According, studies have shown the ability of SA and other natural products in reducing lipid peroxidation and oxidative cellular injuries [14,50]. This bioactivity of SA could be linked with its effective modulation of antioxidant activation including the Nrf2-Keap1 pathway and the HO-1, NQO1, and GCLC genes as previously explained [51]. 

Liver cirrhosis induced by TAA stimulates a series of immunological changes related to the secretion of several inflammatory cytokines (TNF-α and IL-6) leading to further production of reactive oxygen species and free radicals [52]. Pro-inflammatory cytokines such as TNF-α are synthesized mainly by macrophages, catching the neutrophil’s attention to the inflammatory site as it may progressively cause more cellular damage [22]. IL-6 has been also known as a specific pro-inflammatory indicator that can activate the immune system and trigger the anti-inflammation process as shown that IL-6 has the potential the stimulation of hematopoietic stem cells (granulocytes and granulocytes) leading to the initiation of stress response in damaged tissues [53,54]. In contrast, IL-10 has known as an anti-inflammatory cytokine with the potential to lower the inflammatory response and reduce the TNF-α formation. Studies have shown several chemicals can up-regulate the pro-inflammatory and down-regulate the anti-inflammatory cytokines in hepatic and gastric tissues [4,19,55]. Accordingly, the current data have shown that rats exposed to only TAA had higher TNF-α and IL-6 and reduced IL-10 levels than that control rats. While, the SA treatments restored these increases in TNF-α and IL-6 levels and curbed the reduced IL-10 almost to normal levels, indicating the SA potential as an anti-inflammatory agent in cirrhotic rats. The outcome concurs with the previously reported study on the anti-inflammatory actions of SA and other medicinal phytochemicals in chemically-induced cirrhotic rats [4,47]. Similarly, researchers have shown the inhibitory effect of SA against alcoholic liver disease and linked this action with its regulatory actions on transcription factors associated with inflammation and tumor growth [56].

## 5. Conclusions

The current study for the first time, to our best knowledge, showed the curbing efficacy of SA in TAA-induced hepatic fibrotic damage. This bioactivity was correlated with the potentials of SA in the Up-regulation of HSP 70, antioxidant enzymes, and deterring the inflammatory, hepatocyte proliferation and reducing mitotic index as evidenced by its inhibitory potentials of α-SMA expression. The outcomes provide scientific backup for the therapeutic efficacy of SA as a hepatoprotective agent and suggest future investigations on a much larger scale for deep exploration of this compound as a possible new medicinal source of hepatic diseases.

## Figures and Tables

**Figure 1 biomedicines-11-01447-f001:** Effect of SA on sections of liver and kidney tissues from rats under acute toxicity test. (**A**,**B**), control group; (**C**,**D**), rats received 2 g/kg of SA; (**E**,**F**), rats received 4 g/kg of SA. The microscopic observation showed non-significant changes in the tissue composition of the livers and kidneys between the experimented rats and control rats (Hematoxylin and Eosin stain, 20×).

**Figure 2 biomedicines-11-01447-f002:** Histopathological tissue views of the liver. Hematoxylin and Eosin (H & E) and Mason Trichrome stains (MT), and Gross appearance (GA) of liver from (**A**), control group; (**B**), TAA group; (**C**), silymarin group; (**D**), 20 mg/kg SA; (**E**), 40 mg/kg SA. The colored hepatic tissues were observed by Nikon microscope (Y-THS, Japan). 20× magnification.

**Figure 3 biomedicines-11-01447-f003:** Shows score for deposited collagens in liver tissues stained with Mason trachoma. Values indicated significant as ns, non-significant; *, *p* < 0.5; ***, *p* < 0.001; ****, *p* < 0.0001.

**Figure 4 biomedicines-11-01447-f004:** Effect of SA on HSP 70 protein expression in TAA-induced liver cirrhosis in rats. (**A**) Control rats showed normal liver tissue distribution without HSP 70 expression. (**B**) Positive control rats had severe liver damage presented by significantly lower HSP 70 protein expression in TAA-induced cirrhosis in rats (**B**,**F**). (**C**) Silymarin rats experienced mild liver injury and up-regulated the HSP 70 in TAA-induced cirrhosis in rats (**C**,**F**). (**D**) SA 20 mg/kg treated rats had moderate hepatocyte damage and up-regulated the HSP 70 staining in TAA-induced liver cirrhosis in rats (**D**,**F**). (**E**) SA 40 mg/kg treated rats presented mild hepatocellular damage and up-regulated the HSP 70 expression in TAA-induced liver cirrhosis in rats (**E**,**F**). (HSP 70 stain, magnification 100×). The data are presented as means ±  SEM. The antigen site appears as a brown color. ****, mean significant at *p* < 0.0001.

**Figure 5 biomedicines-11-01447-f005:** Effect of SA on Alpha-smooth muscle actin (α-SMA) expression in the liver of TAA induced liver cirrhosis in rats. Control group (**A**,**F**); Positive control rats (**B**,**F**); Silymarin group (**C**,**F**); SA 20 mg/kg group (**D**,**F**); SA 40 mg/kg group (**E**,**F**). The colored liver tissues were observed by Nikon microscope (Y-THS, Japan). 100× magnification. *, *p* < 0.5; **, *p* < 0.01; ***, *p* < 0.001; ****, *p* < 0.0001.

**Figure 6 biomedicines-11-01447-f006:** Effect of SA on the SOD (**a**), CAT (**b**) and MDA (**c**) levels of liver homogenate in TAA-induces liver cirrhosis in rats. Values are presented as mean ± SEM. (*n* = 6 rate/group). ns, non-significant; *, *p* < 0.5; **, *p* < 0.01; ***, *p* < 0.001; ****, *p* < 0.0001.

**Figure 7 biomedicines-11-01447-f007:** Effect of SA on TNF a (**a**), IL-6 (**b**) and IL-10 (**c**) in TAA-induced liver cirrhosis in rats. G1, control group; G2, TAA group; G3, silymarin group; G4, SA 20 mg/kg group; G5, SA 40 mg/kg group. ns, non-significant; *, *p* < 0.5; **, *p* < 0.01; ****, *p* < 0.0001.

**Table 1 biomedicines-11-01447-t001:** (**A**) Acute toxicity: Effect of SA on liver biochemistry in rats. (**B**) Acute toxicity: Effects of SA on kidney biochemistry in rats.

(**A**)
**Animals Groups**	**ALP** **(IU/L)**	**ALT** **(IU/L)**	**AST** **(IU/L)**	**T. Bilirubin** **(µmol/L)**	**T. Protein** **(g/L)**	**Albumin** **(g/L)**
Control10% Tween 20	79 ± 2.34	39.1 ± 3.62	59.2 ± 1.22	1.23 ± 0.23	71.2 ± 2.19	26.2 ± 1.3
SA2 g/kg	71.16 ± 2.10	48.16 ± 2.59	62.23 ± 2.12	1.35 ± 0.03	70.30 ± 2.72	24.54 ± 3.2
SA4 g/kg	75 ± 2.30	38.2 ± 1.76	58 ± 2.15	1.30 ± 0.04	74.25 ± 3.51	25.36 ± 3.30
(**B**)
**Animals Groups**	**Sodium** **mmol/L**	**Potassium** **mmol/L**	**Chloride** **mmol/L**	**Urea** **mmol/L**	**Creatinine** **µmol/L**
Control10% Tween 20	148 ± 2.37	5.1 ± 0.32	107 ± 3.039	4.41 ± 0.21	42.22 ± 3.60
SA2 g/kg	145.21 ± 2.35	5.32 ± 3.65	136.5 ± 3.82	5.28 ± 0.32	37.23 ± 2.72
SA4 g/kg	141.22 ± 3.32	5.1 ± 0.30	100.23 ± 2.41	4.72 ± 0.45	40.32 ± 3.20

(**A**) Data were presented as mean ± SEM. The effects of SA on serum alanine aminotransferase (ALT), aspartate aminotransferase (AST), and alkaline phosphatase (ALP) activities, as well as total bilirubin, albumin, and protein levels. The data are presented as mean ± SE (*n* = 6 per group). There was no significant alteration in the liver biochemical parameters of experimented rats. The significance level was set at *p* < 0.05. (**B**) Data are presented as mean ± SEM. Data showed a lack of statistical changes between the experimented groups. The significance level was set at *p* < 0.05.

**Table 2 biomedicines-11-01447-t002:** Effect of SA on body weight, liver weight, and liver index in TAA-induced liver cirrhosis in rats.

Groups	Body Weight (gm)	Liver Weight(gm)	Liver Index LW/BW %
Negative control	343.14 ± 5.75 ^a^	8.25 ± 0.03 ^a^	2.40
TAA control(200 mg/kg)	210.34 ± 3.30 ^d^	11.28 ± 0.04 ^b^	5.36
Silymarin(50 mg/kg) + TAA(200 mg/kg)	323.20 ± 6.40 ^b^	9.45 ± 0.07 ^a^	2.92
SA(20 mg)/kg + TAA(200 mg/kg)	264.42 ± 4.54 ^c^	10.27 ± 0.04 ^a^	3.88
SA(40 mg/kg) + TAA(200 mg/kg)	334.5 ± 5.33 ^c^	8.9 ± 0.05 ^a^	2.66

Data are shown as mean ± SEM (*n* = 6). Values with different superscripts within the same column are considered significant according to Tukey’s honesty test at *p* < 0.05.

**Table 3 biomedicines-11-01447-t003:** Effects of SA on liver biochemistry in serum of rats intoxicated by TAA.

Groups	ALPIU/L	ALTIU/L	ASTIU/L	T. Bilirubin(µmol/L)	Proteing/L	Albuming/L
Control (10% Tween 20)	73.54 ± 3.30 ^a^	40.3 ± 3.18 ^a^	63.15 ± 2.5 ^a^	11.20 ± 0.19 ^a^	70.14 ± 1.20 ^a^	24.27 ± 1.24 ^a^
TAA control(200 mg/kg)	257.3 ± 3.41 ^d^	177 ± 2.01 ^d^	255.0 ± 3.4 ^d^	6.21 ± 0.03 ^d^	43.30 ± 3.41 ^c^	14.81 ± 2.07 ^e^
Silymarin(50 mg/kg) + TAA(200 mg/kg)	79.45 ± 1.0 ^a^	46.4 ± 1.21 ^a^	69.87 ± 1.6 ^a^	1.42 ± 0.07 ^b^	63.45 ± 1.32 ^b^	21.53 ± 2.04 ^b^
SA (20 mg/kg) + TAA(200 mg/kg)	94.75 ± 1.54 ^c^	84.22 ± 1.3 ^c^	86.42 ± 1.5 ^c^	2.50 ± 0.07 ^c^	59.12 ± 1.2 ^b^	16.56 ± 1.17 ^d^
SA (40 mg/kg) + TAA(200 mg/kg)	81.57 ± 3.23 ^b^	55.55 ± 3.5 ^b^	74.45 ± 2.0 ^b^	1.22 ± 0.7 ^b^	62.22 ± 1.1 ^b^	19.47 ± 0.99 ^c^

Data presented as Mean ± SEM. (*n* = 6). Values with different letters within the same column are considered significant according to Tukey’s honesty test at *p* < 0.05.

## Data Availability

Details regarding the current study are available on request.

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
