# Peer review of "Sinapic Acid Attenuate Liver Injury by Modulating Antioxidant Activity and Inflammatory Cytokines in Thioacetamide-Induced Liver Cirrhosis in Rats"

_biomedicines, 2023, doi:10.3390/biomedicines11051447_

Round 1
Reviewer 1 Report
The manuscript (biomedicines-2380840) is fascinating, within the journal scope, and presents a detailed biological evaluation.
Some issues of format should be correct. See, for example, lines 86-87. The references should be cited in the order. Just to mention an example, reference 24 is the last one of the introduction (line 104), and reference 25 appears in line 385 after all the other references. In my opinion, some of the self-citations are not necessary.
The statistical part is short; I think the obtained data need more statistical analysis.
The conclusions also need some improvements. Why use sinapic acid? Ok, it is a natural compound found in food! But is the amount sufficient for suggesting ingesting that type of food, or must sinapic acid be obtained by synthesis? The authors did not isolate the compound; they did purchase it. So the idea is to substitute synthetic compounds with another synthetic compound, and the only difference is that the last one can be found in currently ingested food. The authors should convince the readers that it is more important to use natural compounds. The last paragraph of the introduction is not sufficient. Additionally, the conclusions do not give extra information.
Author Response
Reviewer 1
The manuscript (biomedicines-2380840) is fascinating, within the journal scope, and presents a detailed biological evaluation.
Some issues of format should be correct. See, for example, lines 86-87. The references should be cited in the order. Just to mention an example, reference 24 is the last one of the introduction (line 104), and reference 25 appears in line 385 after all the other references. In my opinion, some of the self-citations are not necessary.
-Response, the sections have been reordered, and the references are now in the right order.
-Response, unnecessary self -citations has been removed.
The statistical part is short; I think the obtained data need more statistical analysis.
-Response, this section has been rewritten.
The conclusions also need some improvements. Why use sinapic acid? Ok, it is a natural compound found in food! But is the amount sufficient for suggesting ingesting that type of food, or must sinapic acid be obtained by synthesis? The authors did not isolate the compound; they did purchase it. So the idea is to substitute synthetic compounds with another synthetic compound, and the only difference is that the last one can be found in currently ingested food. The authors should convince the readers that it is more important to use natural compounds.
-Response, the current study explores hepatoprotective effects of sinapic acid in TAA-induced liver cirrhosis for the first time. SA compound has been explored for various bioactivities and have shown therapeutic efficiency against wide variety of diseases.
The SA has been explored as a potential new therapeutic (drug) source against liver cirrhosis. Only 20 mg/kg and 40 mg/kg were enough to reduce the hepatic tissue damage induced by TAA.
-The current study explored SA as potential new source of hepatoprotection agent and found significant inhibitory efficacy of this compound against suggested TAA-induced liver cirrhosis in rats. The purpose is to determine the medicinal efficacy of SA in against liver cirrhosis, which only small amount of 20 mg/kg and 40 mg/kg were enough to reduce the hepatic tissue damage induced by TAA.
-The current study showed the hepatoprotective roles of SA against TAA-induced liver cirrhosis in rats in a two-month trial. The outcomes showed significant efficacy of SA in reducing hepatic injury induced by TAA, however, the livers were of SA-treated rats had significantly lower tissue damages in mucous and sub-mucosal layers compared to positive control. The outcomes provide scientific backup for the therapeutic efficacy of SA as hepatoprotive agent and suggests future investigations on much larger scale for deep exploration of this compound as possible new medicinal source of hepatic diseases.
-The idea is to attract more attentions on the therapeutic efficacy of this natural compound, which can be detected and extracted from various natural sources and plant species.
The last paragraph of the introduction is not sufficient.
-The last paragraph of introduction has been rewritten.
-The statistical section has been rewritten.
Additionally, the conclusions do not give extra information.
-The conclusion section added with more information and suggestions.
Reviewer 2 Report
The authors describe changes in liver function and histopathological profile caused by the natural product sinapic acid in an in vivo model of liver fibrosis. However, since the hepatoprotective effects of sinapic acid were previously reported, the results of the study are predictable and of low significance. The manuscript does not provide any new mechanistic information on the cellular targets of sinapic acid.
The others should cite and discuss some additional publications of relevance e.g. PMID: 34149421 and 36235257 .
Minor proofreading will improve the English quality.
Author Response
Reviewer 2
Reviewer 2
The authors describe changes in liver function and histopathological profile caused by the natural product sinapic acid in an in vivo model of liver fibrosis. However, since the hepatoprotective effects of sinapic acid were previously reported, the results of the study are predictable and of low significance. The manuscript does not provide any new mechanistic information on the cellular targets of sinapic acid.
- Systematic search showed lack of any concise experimental analysis explaining the bio-mechanisms behind the protective role of SA against liver cirrhosis. Therefore, the current study aims to investigate the hepatoprotection efficiency of SA in TAA-induced liver cirrhosis in rats for the first time.
The others should cite and discuss some additional publications of relevance, e.g. PMID: 34149421 and 36235257.
-Response, those references were added accordingly.
Reviewer 3 Report
The article written by Zaenah Zuhair Alamri et al. presents studies on the ability of sinapic acid to alleviate thioacetamide-induced liver damage in rats. The manuscript can be accepted for publication after minor revision. The authors should revise the manuscript according to the following comments.
1. line 109. In the first sentence "Rats treated with 20 and 40 mg/kg were observed for 2 weeks". Please add what they were treated with?
2. Table 2 and Table 3. Please add to the table what amounts of TAA were administered.
3. Line 140. Is the reduction in liver weight when administered with SA a positive or negative effect?
4. Did you also study how did the volume of the liver change?
5. Does SA remove previous liver damage?
Minor editing of English language required.
Author Response
Reviewer 3
The article written by Zaenah Zuhair Alamri et al. presents studies on the ability of sinapic acid to alleviate thioacetamide-induced liver damage in rats. The manuscript can be accepted for publication after minor revision. The authors should revise the manuscript according to the following comments.
- line 109. In the first sentence "Rats treated with 20 and 40 mg/kg were observed for 2 weeks". Please add what they were treated with?
In response, this issue is resolved accordingly.
- Table 2 and Table 3. Please add to the table what amounts of TAA were administered.
Response, this issue is resolved accordingly and doses of TAA were added.
- Line 140. Is the reduction in liver weight when administered with SA a positive or negative effect?
Response, studies have reported the TAA efficacy in changing the weightage of the body and bulky liver formation (increased liver weight) in various rat trials [4,42,43]. These body/liver weightage modulations were almost normalized in SA-treated rats, which could be correlated with the dyslipidemia[44], anti-inflammatory[45], antioxidant[46] bioactivities of SA as reported by previous researchers. Similarly, researchers have reported the SA efficacy in restoring the body and liver weightage to normal in dimethylnitrosamine (DMN)-induced chronic liver damage in rats [47].
- Did you also study how did the volume of the liver change?
In response, the current study did not measure the volume of obtained liver from various rat groups. However, data analysis showed higher Liver Index (LW/BW %) for TAA-treated rats (5.36%). While, lower values (3.88% and 2.66%) were shown for rats treated with SA 20 mg/kg and 40 mg/kg, respectively.
- Does SA remove previous liver damage?
-The current study showed the hepatoprotective roles of SA against TAA-induced liver cirrhosis in rats in a two-month trial. The outcomes showed significant efficacy of SA in reducing hepatic injury induced by TAA, however, the livers were of SA-treated rats had significantly lower tissue damages in mucous and sub-mucosal layers compared to positive control. The outcomes provide scientific backup for the therapeutic efficacy of SA as hepatoprotective agent and suggests future investigations on much larger scale for deep exploration of this compound as possible new medicinal source of hepatic diseases.